# SEMI-SUPERVISED COUNTERFACTUAL EXPLANATIONS

## ABSTRACT

Counterfactual explanations for machine learning models are used to find minimal interventions to the feature values such that the model changes the prediction to a different output or a target output. A valid counterfactual explanation should have likely feature values. Here, we address the challenge of generating counterfactual explanations that lie in the same data distribution as that of the training data and more importantly, they belong to the target class distribution. This requirement has been addressed through the incorporation of auto-encoder reconstruction loss in the counterfactual search process. Connecting the output behavior of the classifier to the latent space of the auto-encoder has further improved the speed of the counterfactual search process and the interpretability of the resulting counterfactual explanations. Continuing this line of research, we show further improvement in the interpretability of counterfactual explanations when the auto-encoder is trained in a semi-supervised fashion with class tagged input data. We empirically evaluate our approach on several datasets and show considerable improvement in-terms of several metrics.

## 1 INTRODUCTION

Recently counterfactual explanations have gained popularity as tools of explainability for AI-enabled systems. A counterfactual explanation of a prediction describes the smallest change to the feature values that changes the prediction to a predefined output. A counterfactual explanation usually takes the form of a statement like, "You were denied a loan because your annual income was $30,000$. If your income had been $45,000$, you would have been offered a loan ". Counterfactual explanations are important in the context of AI-based decision-making systems because they provide the data subjects with meaningful explanations for a given decision and the necessary actions to receive a more favorable/desired decision in the future. Application of counterfactual explanations in the areas of financial risk mitigation, medical diagnosis, criminal profiling, and other sensitive socio-economic sectors is increasing and is highly desirable for bias reduction.

Apart from the challenges of sparsity, feasibility, and actionability, the primary challenge for counterfactual explanations is their interpretability. Higher levels of interpretability lead to higher adoption of AI-enabled decision-making systems. Higher values of interpretability will improve the trust amongst data subjects on AI-enabled decisions. AI models used for decision making are typically black-box models, the reasons can either be the computational and mathematical complexities associated with the model or the proprietary nature of the technology. In this paper, we address the challenge of generating counterfactual explanations that are more likely and interpretable. A counterfactual explanation is interpretable if it lies within or close to the model's training data distribution. This problem has been addressed by constraining the search for counterfactuals to lie in the training data distribution. This has been achieved by incorporating an auto-encoder reconstruction loss in the counterfactual search process. However, adhering to training data distribution is not sufficient for the counterfactual explanation to be likely. The counterfactual explanation should also belong to the feature distribution of its target class. To understand this, let us consider an example of predicting the risk of diabetes in individuals as high or low. A sparse counterfactual explanation to reduce the risk of diabetes might suggest a decrease in the body mass index (BMI) level for an individual while leaving other features unchanged. The model might predict a low risk based on this change and the features of this individual might still be in the data distribution of the model. However, they will not lie in the data distribution of individuals with low risk of diabetes because of other relevant features of low-risk individuals like glucose tolerance, serum insulin, diabetes pedigree, etc.

To address this issue, authors in Van Looveren & Klaise (2019) proposed to connect the output behavior of the classifier to the latent space of the auto-encoder using prototypes. These prototypes guide the counterfactual search process in the latent space and improve the interpretability of the resulting counterfactual explanations. However, the auto-encoder latent space is still unaware of the class tag information. This is highly undesirable, especially when using a prototype guided search for counterfactual explanations on the latent space. In this paper, we propose to build a latent space that is aware of the class tag information through joint training of the auto-encoder and the classifier. Thus the counterfactual explanations generated will not only be faithful to the entire training data distribution but also faithful to the data distribution of the target class.

We show that there are considerable improvements in interpretability, sparsity and proximity metrics can be achieved simultaneously, if the auto-encoder trained in a semi-supervised fashion with class tagged input data. Our approach does not rely on the availability of train data used for the black box classifier. It can be easily generalized to a post-hoc explanation method using the semi-supervised learning framework, which relies only on the predictions on the black box model. In the next section we present the related work. Then, in section 3 we present preliminary definitions and approaches necessary to introduce our approach. In section 4, we present our approach and empirically evaluate it in section 5.

## 2 RELATED WORK

Counterfactual analysis is a concept derived from from causal intervention analysis. Counterfactuals refer to model outputs corresponding to certain imaginary scenarios that we have not observed or cannot observe. Recently Wachter et al. (2017) proposed the idea of model agnostic (without opening the black box) counterfactual explanations, through simultaneous minimization of the error between model prediction and the desired counterfactual and distance between original instance and their corresponding counterfactual. This idea has been extended for multiple scenarios by Mahajan et al. (2019), Ustun et al. (2019) , Poyiadzi et al. (2020) based on the incorporation of feasibility constraints, actionability and diversity of counterfactuals. Authors in Mothilal et al. (2020) proposed a framework for generating diverse set of counterfactual explanations based on determinantal point processes. They argue that a wide range of suggested changes along with a proximity to the original input improves the chances those changes being adopted by data subjects. Causal constraints of our society do not allow the data subjects to reduce their age while increasing their educational qualifications. Such feasibility constraints were addressed by Mahajan et al. (2019) and Joshi et al. (2019) using a causal framework. Authors in Mahajan et al. (2019) addresses the feasibility of counterfactual explanations through causal relationship constraints amongst input features. They present a method that uses structural causal models to generate actionable counterfactuals. Authors in Joshi et al. (2019) propose to characterize data manifold and then provide an optimization framework to search for actionable counterfactual explanation on the data manifold via its latent representation. Authors in Poyiadzi et al. (2020) address the issues of feasibility and actionability through feasible paths, which are based on the shortest path distances defined via density-weighted metrics.

An important aspect of counterfactual explanations is their interpretability. A counterfactual explanation is more interpretable if it lies within or close to the data distribution of the training data of the black box classifier. To address this issue Dhurandhar et al. (2018) proposed the use of auto-encoders to generate counterfactual explanations which are "close" to the data manifold. They proposed incorporation of an auto-encoder reconstruction loss in counterfactual search process to penalize counterfactual which are not true to the data manifold. This line of research was further extended by Van Looveren & Klaise (2019), they proposed to connect the output behaviour of the classifier to the latent space of the auto-encoder using prototypes. These prototypes improved speed of counterfactual search process and the interpretability of the resulting counterfactual explanations.

While Van Looveren & Klaise (2019) connects the output behaviour of the classifier to the latent space through prototypes, the latent space is still unaware of the class tag information. We propose to build a latent space which is aware of the class tag information through joint training of the auto-encoder and the classifier. Thus the counterfactual explanations generated will not only be faithful the entire training data distribution but also faithful the data distribution of the target class. In a post-hoc scenario where access to the training data is not guaranteed, we propose to use the input-output pair data of the black box classifier to jointly train the auto-encoder and classifier in the semi-supervised learning framework. Authors in Zhai & Zhang (2016), Gogna et al. (2016) have explored the use semi-supervised auto-encoders for sentiment analysis and analysis of biomedical signal analysis. Authors

in Haiyan et al. (2015) propose a joint framework of representation and supervised learning which guarantees not only the semantics of the original data from representation learning but also fit the training data well via supervised learning. However, as far as our knowledge goes, semi-supervised learning has not been used to generate counterfactual explanations and we experimentally show that semi-supervised learning framework generates more interpretable counterfactual explanations.

## 3 PRELIMINARIES

Let $\mathcal{D} = \{\mathbf{x}_i, y_i\}_{i=1\ldots N}$ be the supervised data set where $\mathbf{x}_i \in \mathcal{X}$ is $d$-dimensional input feature space for a classifier and $y_i \in \mathcal{Y} = \{1, 2, \ldots, \ell\}$ is the set of outputs for a classifier. Throughout this paper we assume the existence of a black box classifier $h : \mathcal{X} \to \mathcal{Y}$ trained on $\mathcal{D}$ such that $\hat{y} = h(\mathbf{x}) = \arg\max_{c \in \mathcal{Y}} p(y = c \mid \mathbf{x}, \mathcal{D})$ where $p(y = c \mid \mathbf{x}, \mathcal{D})$ is prediction score/probability for class $c$ with an input $\mathbf{x}$. Based on Wachter et al. (2017), counterfactual explanations can be generated by trading off between prediction loss and sparsity. This is achieved by optimizing a linear combination of the prediction loss ($L_{pred}$) and loss of sparsity ($L_{sparsity}$) as $L = c \cdot L_{pred} + L_{sparsity}$. Prediction loss typically measures the distance between current prediction and the target class, whereas sparsity loss function measures the perturbation from the initial instance $\mathbf{x}_0$ with class tag $t_0$. This approach generates counterfactual explanations which can reach their target class with a sparse perturbation to the initial instance. However, they need not necessarily respect the input data distribution of the classifier, hence, resulting in unreasonable values for $\mathbf{x}^{cfe}$.

Authors in Dhurandhar et al. (2018) addressed this issue through incorporation of $L_2$ reconstruction error for $\mathbf{x}^{cfe}$ evaluated through an autoencoder (AE) trained on the input data $\mathcal{X}$ as $L_{recon}^{\mathcal{X}}(\mathbf{x}) = \|\mathbf{x} - AE_{\mathcal{X}}(\mathbf{x})\|_2^2$ where $AE_{\mathcal{X}}$ represents the auto-encoder trained on entire training dataset $\mathcal{X}$. The auto-encoder loss function $L_{recon}^{\mathcal{X}}$ penalizes counterfactual explanations which do not lie within the data-distribution. However, Van Looveren & Klaise (2019) illustrated that incorporating $L_{recon}^{\mathcal{X}}$ in $L$ may result in counterfactual explanations which lie inside the input data-distribution but they may not be interpretable. To this end, Van Looveren & Klaise (2019) proposes addition of a prototype loss function $L_{proto}^{\mathcal{X}}$ to $L$ to make $\mathbf{x}^{cfe}$ more interpretable and improve the counterfactual search process through prototypes in the latent space of auto-encoder. $L_{proto}^{\mathcal{X}}$ is the $L_2$ error between the latent encoding of $\mathbf{x}$ and cluster centroid of the target class in the latent space of the encoder defined as $\text{proto}_t$ (short for target prototype) as $L_{proto}^{\mathcal{X}}(\mathbf{x}, \text{proto}_t) = \|ENC_{\mathcal{X}}(\mathbf{x}) - \text{proto}_t\|_2^2$, where $ENC_{\mathcal{X}}$ represents encoder part of the auto-encoder $AE_{\mathcal{X}}$ and $ENC_{\mathcal{X}}(\mathbf{x})$ represents the projection of $\mathbf{x}$ on to the latent space of the auto-encoder. Given a target $t$, the corresponding $\text{proto}_t$ can be defined as

$$\text{proto}_t = \frac{1}{K} \sum_{k=1}^{K} ENC_{\mathcal{X}}(\mathbf{x}_k^t) \tag{1}$$

where $\mathbf{x}_k^t$ represent the input instances corresponding to the class $t$ such that $\{ENC_{\mathcal{X}}(\mathbf{x}_k^t)\}_{k=1,\ldots,K}$ are the $K$ nearest neighbors of $ENC_{\mathcal{X}}(\mathbf{x}_0)$. For applications where target class $t$ is not pre-defined, a suitable replacement for $\text{proto}_t$ is evaluated by finding the nearest prototype $\text{proto}_j$ of class $j \neq t_0$ to the encoding of $\mathbf{x}_0$, given by $j = \arg\min_{i \neq t_0} \|ENC_{\mathcal{X}}(\mathbf{x}_0) - \text{proto}_i\|_2$. Then prototype loss $L_{proto}$ can be defined as $L_{proto}(\mathbf{x}, \text{proto}_j) = \|ENC_{\mathcal{X}}(\mathbf{x}_0) - \text{proto}_j\|_2^2$. According to Van Looveren & Klaise (2019) the loss function $L_{proto}$ explicitly guides the encoding of the counterfactual explanation to the target prototype (or the nearest protoytpe $\text{proto}_{i \neq t_0}$). Thus we have a loss function $L$ given by

$$L = c \cdot L_{pred} + L_{sparsity} + \gamma \cdot L_{recon}^{\mathcal{X}} + \theta \cdot L_{proto}^{\mathcal{X}} \tag{2}$$

where $c$, $\gamma$ and $\theta$ are hyper-parameters tuned globally for each data set. For detailed descriptions of these parameters and their impact on the counterfactual search, we refer the readers to Ltd. In this paper, we propose an alternate version of this loss function. The constituent loss functions $L_{recon}^{\mathcal{X}}$ and $L_{proto}^{\mathcal{X}}$ are based on an auto-encoder trained in an unsupervised fashion. In the next section we motivate the use of an auto-encoder trained using class tagged data in a semi-supervised fashion.

## 4 SEMI-SUPERVISED COUNTERFACTUAL EXPLANATIONS

(a) *Classification*  (b) *Autoencoder*  (c) *Jointly trained model*

For the supervised classification data set $\mathcal{D} = \langle \mathcal{X}, \mathcal{Y} \rangle$ machine learning methods learn a classifier model $h : \mathcal{X} \mapsto \mathcal{Y}$ (see figure 1a). This process of learning involves minimizing a loss function of the form: $\mathcal{E}_{entropy} = -\sum_{j}^{\ell} \sum_{i}^{N} \mathbb{I}(y_i = j) * \log(p(y = j \mid \mathbf{x}_i, \mathcal{D}))$. Auto-encoder is a neural network framework which learns a latent space representation $\mathbf{z} \in \mathcal{Z}$ for input data $\mathbf{x} \in \mathcal{X}$ along with an invertible mapping ($\phi_{\mathcal{X}}^{-1}$) (see figure 1b) in an unsupervised fashion. The subscript $\mathcal{X}$ represents the unsupervised training of $\phi_{\mathcal{X}}$ and $\phi_{\mathcal{X}}^{-1}$ only on the dataset $\mathcal{X}$. The un-supervised learning framework tries to learn data compression using continuous map $\phi_{\mathcal{X}}$, while minimizing the reconstruction loss: $\mathcal{E}_{autoenc} = \sqrt{\frac{1}{N} \sum_{i}^{N} |\mathbf{x}_i - \mathbf{x}_i'|^2}$. In this paper we consider only undercomplete autoencoders that produce a lower dimension representation ($\mathcal{Z}$) of an high dimensional space ($\mathcal{X}$), while the decoder network ensures the reconstruction guarantee ($\mathbf{x} \approx \phi^{-1}(\phi(\mathbf{x}))$. A traditional undercomplete auto-encoder captures the correlation between the input features for the dimension reduction.

## 4.1 Joint training: semi-supervised learning

For the purpose of generating counterfactual explanations, we propose to use a generic neural architecture for an undercomplete auto-encoder jointly trained with the classifier model (figure 1c). The proposed system would be trained with a joint loss, defined as a linear combination of cross-entropy loss and reconstruction loss $\mathcal{E}_{joint} = w_1 \cdot \mathcal{E}_{entropy} + w_2 \cdot \mathcal{E}_{autoenc}$. This architecture relies on the class tag information $y_i$ for every input $\mathbf{x}_i$ used to train the auto-encoder. The subscript $\mathcal{D}$ in figure 1c represents the training of $\phi_{\mathcal{D}}$ and $\phi_{\mathcal{D}}^{-1}$ using dataset $\mathcal{X}$ tagged with classes from $\mathcal{Y}$ in the spirit of semi-supervised learning. This jointly trained the auto-encoder and the corresponding encoder will be represented by $AE_{\mathcal{D}}$ and $ENC_{\mathcal{D}}$. This generic architecture can be implemented in multiple ways based on selection of classifier model $h$, architecture of neural network for $\phi_{\mathcal{D}}$ and weights $w_1$ and $w_2$. Also, if the entire supervised $\mathcal{D}$ is unavailable and the class tag information is available only for $\mathcal{D}_t = \{\mathbf{x}_i, y_i\}$ where $i = 1, \dots, m < n$ and un-tagged data is available as $\mathcal{D}_u = \{\mathbf{x}_i\}$ where $i = m + 1, \dots, n$ then the joint training approach presented in figure 1c can be generalized to semi-supervised learning framework.

## 4.2 Characterization of the latent embedding space

The lower dimension representation (embedding) $\mathbf{z} \in \mathcal{Z}$ obtained through joint training using $\mathcal{D} = \langle \mathcal{X}, \mathcal{Y} \rangle$ is relatively more richer than its unsupervised counterpart trained using $\mathcal{X}$. Thus, we claim that the counterfactual explanations generated through auto-encoders trained on class tagged data will not only be more faithful to the training data distribution but also be more faithful to the target class data distribution and hence more interpretable. We evaluate this claim by generating counterfactual evaluations on several data sets and compare them through suitable metrics. However, to illustrate the motivation behind the proposed approach we consider the German credit data set[1]. This data set consists of credit risk for over 1000 individuals. It contains 13 categorical and 7 continuous features. The target variable is a binary decision whether borrower will be a defaulter (`high risk`) or not (`low risk`). The counterfactual explanations in this case would typically be the necessary feature changes to make an individual `low risk`. In figure 2, we try to characterize and visualize the embedding space in two dimensions ($\mathcal{Z} \subseteq \mathbb{R}^2$) for the German data set. We plot classifier outputs and classification probability contours in the latent embedding space $\mathbf{z} \in \mathcal{Z}$ for the separately trained auto-encoder $AE_{\mathcal{X}}$ (henceforth termed as unsupervised auto-encoder) in figures 2a and 2c and the jointly trained auto-encoder $AE_{\mathcal{D}}$ (henceforth termed as semi-supervised auto-encoder) in figures 2b and 2d.

For a fair comparison between unsupervised and semi-supervised frameworks, we use the same neural network framework for both $\phi_{\mathcal{X}}$ and $\phi_{\mathcal{D}}$. It can be observed that the unsupervised auto-encoder produces an embedding that does not clearly separate `high risk` and `low risk` classes (figure 2a). In the embedding space, the classification probability contours also overlap significantly (figure 2c) for unsupervised case. For the semi-supervised auto-encoder, the embeddings and their classification probability contours are clearly more separated than their unsupervised counterparts (figures 2b and 2d). Clearly the separation of class clusters is due to inclusion of classification task in the auto-encoder training process. The resulting embedding is indicative of distinct distribution of features between `low risk` and `high risk` individuals. Thus, semi-supervised framework learns highly discriminative embeddings and can be utilized to better counterfactual explanations. To do so, we re-define the loss function $L$ for counterfactual explanation as:

$$L = c \cdot L_{pred} + L_{sparsity} + \gamma \cdot L_{recon}^{\mathcal{D}} + \theta \cdot L_{proto}^{\mathcal{D}} \tag{3}$$

---

[1] https://archive.ics.uci.edu/ml/datasets/statlog+(german+credit+data)

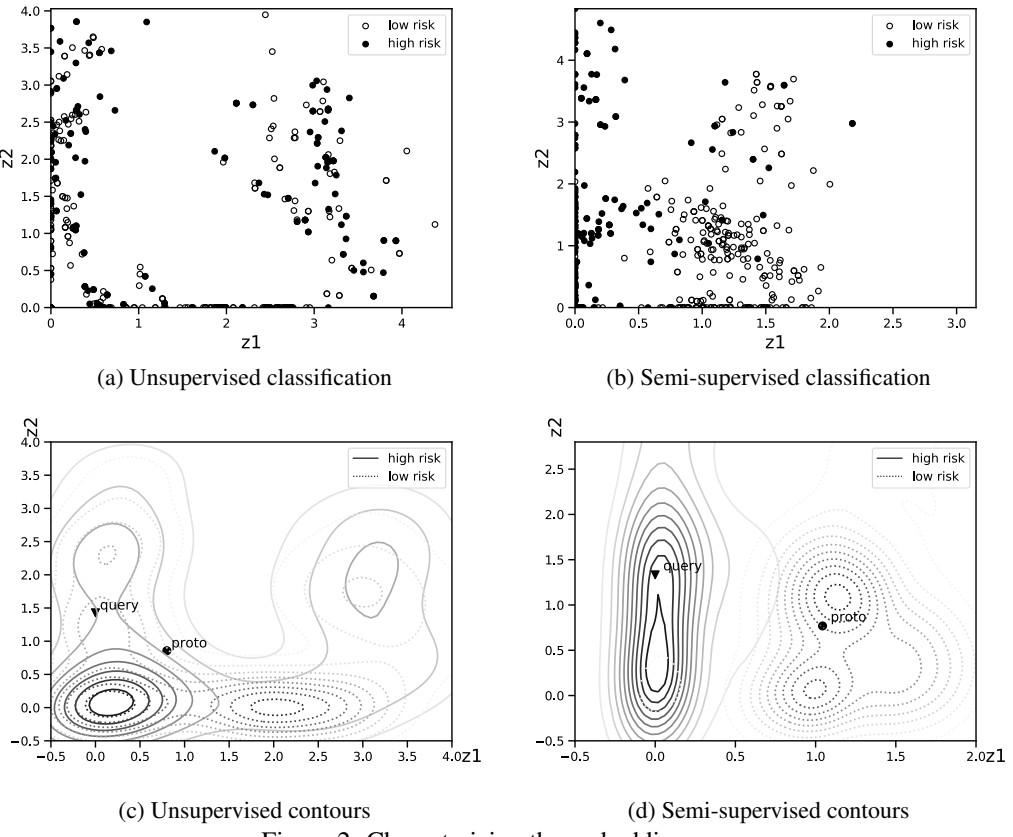

(a) Unsupervised classification          (b) Semi-supervised classification

(c) Unsupervised contours          (d) Semi-supervised contours

Figure 2: Characterizing the embedding space

To illustrate the effect of $L_{proto}^{\mathcal{D}}$ in comparison with $L_{proto}^{\mathcal{X}}$ on the counterfactual generation process, we consider a data instance in $\mathcal{X}$ (the feature space of the German data set) and we term its corresponding encoding as query $\in \mathcal{Z}$. The class tag associated with query is high risk, hence, the target class for counterfactual explanation is low risk. Based on equation 1, we evaluated the proto associated with the class tag low risk and plotted it for both unsupervised (figure 2c) and semi-supervised (figure 2d) scenarios. It can be observed that proto for the semi-supervised scenario falls clearly in the low risk cluster as opposed to proto for the unsupervised. Under such scenarios, the counterfactual explanations generated will be more interpretable because the uncertainty associated with the class tag of the counterfactual is minimized. Apart from higher interpretability, reduction of uncertainty leads to more robust counterfactual explanations. Because a clear separation between clusters will result in a faster counterfactual search towards target proto without meandering towards other classes. In the next section, we experimentally evaluate and compare unsupervised and semi-supervised approaches to generate counterfactual explanations.

## 5 EXPERIMENTS AND RESULTS

We show that semi-supervised training has the potential to generate more interpretable counterfactual explanations. However, interpretability is a function of several parameters of the joint training framework, parameters used for joint optimization of loss functions in the counterfactual search process, parameters of neural networks used for auto-encoders, and parameters of the models used for classification. Hence, we use a common neural network architecture for both unsupervised and semi-supervised frameworks during the comparison. To handle categorical data, we generate a categorical embedding based on the architecture proposed by López-Sánchez et al. (2018) and a corresponding decoding layer for the reverse lookup in our framework. We train the categorical embedding layer to achieve the data distribution faithful embedding definition. The embedding converts the categorical variable to continuous vector space, which is consumed into our auto-encoder (unsupervised and semi-supervised) frameworks.

## 5.1 Datasets used for experiments

For evaluation, we consider German credit data set, adult census data set and breast cancer Wisconsin (diagnostic) data set from Dua & Graff (2017), MNIST data set from LeCun & Cortes (2010), COMPAS data set from Larson et al. (2016) and PIMA dataset from Database (2016).

**German credit data set:** Predict an individual's credit risk using 20 distinct features. The data-set contains over 1000 entries. This data-set contains 13 categorical and 7 continuous features. The target variable is a binary decision whether the borrower will be a defaulter or not.

**Adult census data[2]:** This is a multivariate data-set based on census data, with 12 distinct features. Each feature represents an individual from the census data, where the target variable is binary label representing whether the individual income exceeds \$ 50K/yr. The data-set has over 45, 000 entries. In this study, we have ignored the entries with missing values.

**MNIST data set [3]:** It is a database of $28 \times 28$ hand-written digit images with 60, 000 training, and 10, 000 test examples, with digit label. In this current work, we posed the recognition problem as a binary decision task of detecting odd or even digit.

**Compas[4]:** COMPAS is a commercial algorithm used by judges and parole officers for scoring a criminal defendant's likelihood of reoffending. The dataset contains over 10, 000 criminal defendants records in Broward County, Florida, and all 10 features considered by the COMPAS algorithm.

**PIMA[5]:** This dataset is made available from the National Institute of Diabetes, Digestive, and Kidney Diseases. The objective of the dataset is to diagnostically predict whether a patient has diabetes or not. The dataset consists of several medical predictor variables, viz. BMI, insulin level, age, and so on, and one binary target variable suggesting whether the data is from a diabetes patient or not. All the features of this data-set are continuous in nature.

**Breast Cancer Winsconsin(Diagonostic) Dataset[6]:** It contains features computed from the digitized images of fine needle aspirate (FNA) of breast masses. All features are continuous in nature. The objective of the dataset is to diagnostically predict whether the mass is malignant or benign.

## 5.2 Experimental setup and dependencies

All experiments are performed on a Linux machine running on a single core, 32 threads, Intel(R) Xeon(R) Gold 6130 @2.10GHz processor with 256GB RAM. For each data-set, an unsupervised auto-encoder model, and one semi-supervised auto-encoder models are trained. These models shared the same neural architecture but trained with different loss functions. The unsupervised auto-encoder is trained with reconstruction loss only, while for the semi-supervised scenario the loss function is defined as a linear combination of reconstruction loss, and classification loss. We use Alibi Klaise et al. to generate counterfactual explanations for both unsupervised and semi-supervised scenarios. Keras Chollet et al. (2015) library is used to build the corresponding auto-encoders frameworks.

## 5.3 Counterfactual evaluation metrics

For a given query instance $\mathbf{x}_q$ the counterfactual explanation is $\mathbf{x}_q^{cfe}$. We used proximity, sparsity, and interpretability losses from Mothilal et al. (2020); Van Looveren & Klaise (2019) as the evaluation metrics for the counterfactual explanations.

**Proximity**: This metric evaluates the distance between the query point and the counterfactual explanation. The proximity is handled separately for continuous and categorical fields.

$$\text{CONT-PROXIMITY} = \frac{1}{k}\sum_{i=1}^{k}\frac{|x_{q,i}^{cfe} - x_{q,i}|}{MAD_i}, \quad \text{CAT-PROXIMITY} = 1 - \frac{1}{k}\sum_{i=1}^{k}I(x_{q,i}^{cfe} \neq x_{q,i})$$

The measure of categorical proximity is normalized between $[0, 1]$, representing a fraction of categorical variables that need to be changed to reach the counterfactual explanation. The continuous proximity is normalized by the median absolute deviation (MAD).

---

[2] https://archive.ics.uci.edu/ml/datasets/adult
[3] http://yann.lecun.com/exdb/mnist/
[4] https://github.com/propublica/compas-analysis
[5] https://www.kaggle.com/uciml/pima-indians-diabetes-database
[6] https://archive.ics.uci.edu/ml/datasets/Breast+Cancer+Wisconsin+(Diagnostic)

**Sparsity**: Sparsity metric reports the fraction of features changed between the query ($\mathbf{x}_q$), and the counterfactual explanation ($\mathbf{x}_q^{cfe}$) and it is defined as $\text{SPARSITY} = 1 - \frac{1}{k}\sum_{i=1}^{k} I(x_{q,i}^{cfe} \neq x_{q,i})$. The sparsity is uniformly defined over categorical and continuous features. A good counterfactual explanation desired to have higher sparsity value.

**Interpretability**: We have used two metrics $IM_1$, and $IM_2$ proposed by Van Looveren & Klaise (2019) to evaluate interpretability. Both these metrics use class-specific auto-encoders to estimate how much the counterfactual explanation conforms to the new class tag distribution. $IM_1$ measures relative reconstruction error of target class over query class, while $IM_2$ measures relative improvement in reconstruction error of the target class over nonclass specific reconstruction error. If $AE_{\mathcal{X}}$ represents the auto-encoder trained on the entire data set $\mathcal{X}$, and $AE_i$ is class-specific autoencoder for the class $i$. Then the two metrics are described as

$$IM_1 = \frac{||\mathbf{x}_q^{cfe} - AE_t(\mathbf{x}_q^{cfe})||_2^2}{||\mathbf{x}_q^{cfe} - AE_{t_0}(\mathbf{x}_q^{cfe})||_2^2 + \epsilon}, \quad IM_2 = \frac{||AE_t(\mathbf{x}_q^{cfe}) - AE_{\mathcal{X}}(\mathbf{x}_q^{cfe})||_2^2}{||\mathbf{x}_q^{cfe}||_1 + \epsilon}$$

| Dataset | | S | | $P_{cat}$ | | $P_{cont}$ | | IM1 | | IM2 | |
|---|---|---|---|---|---|---|---|---|---|---|---|
| | | SS | U | SS | U | SS | U | SS | U | SS | U |
| German Credit | $\mu$ | 0.60 | 0.65 | 0.99 | 0.96 | 0.11 | 0.11 | 0.94 | 0.94 | 0.10 | 0.11 |
| | $\sigma$ | 0.05 | 0.04 | 0.02 | 0.05 | 0.11 | 0.09 | 0.06 | 0.06 | 0.03 | 0.03 |
| COMPAS | $\mu$ | 0.72 | 0.64 | 0.89 | 0.82 | 1.34 | 1.76 | 2.18 | 2.97 | 0.43 | 0.46 |
| | $\sigma$ | 0.20 | 0.27 | 0.31 | 0.38 | 1.05 | 1.52 | 3.43 | 4.45 | 0.41 | 0.39 |
| Adult Income | $\mu$ | 0.88 | 0.85 | 1.00 | 1.00 | 0.12 | 0.15 | 1.30 | 1.32 | 0.07 | 0.07 |
| | $\sigma$ | 0.04 | 0.09 | 0.00 | 0.00 | 0.01 | 0.01 | 0.46 | 0.48 | 0.05 | 0.05 |
| PIMA | $\mu$ | 0.66 | 0.59 | $--$ | $--$ | 0.32 | 0.42 | 1.43 | 1.36 | 0.37 | 0.39 |
| | $\sigma$ | 0.20 | 0.26 | $--$ | $--$ | 0.51 | 0.58 | 0.74 | 0.65 | 0.44 | 0.46 |
| Cancer | $\mu$ | 0.61 | 0.57 | $--$ | $--$ | 0.29 | 0.44 | 1.43 | 1.16 | 0.10 | 0.35 |
| | $\sigma$ | 0.07 | 0.11 | $--$ | $--$ | 0.10 | 0.12 | 0.44 | 0.46 | 0.02 | 0.04 |

Table 1: The top header represents different metrics used for comparison, viz. Sparsity(**S**), categorical proximity ($P_{cat}$), continuous proximity ($P_{cont}$), interpretability metric 1 ($IM_1$), and interpretability metric 2 ($IM_2$). Each metric is paired with two columns: **SS** representing the metric obtained using semi-supervised embedding and **U** representing the metric obtained using classical undercomplete autoencoder.

## 5.4 RESULTS

We have sampled over 100 instances from each data set and generated the corresponding counterfactual explanations to compare unsupervised (U) and semi-supervised (SS) frameworks using the sparsity, proximity, and interpretability metrics as shown in table 1. A higher value of the sparsity metric indicates that the counterfactual explanation has been obtained by perturbing a lesser number of features. Across various data-sets, the counterfactual explanations produced by the (SS) framework are sparser than the solution produced by (U) framework. (SS) framework fares better even for categorical proximity and continuous proximity metrics across all datasets except COMPAS dataset, where continuous proximity of (SS) framework is higher. The interpretability metric $IM_1$, compares the reconstruction loss between query class distribution to target class distribution. The comparison yields an improvement in the (SS) framework for adult income and COMPAS data sets, suggesting the counterfactual explanation produced by a semi-supervised embedding framework is better explained by the target class distribution. For other datasets $IM_1$ is either same for both the frameworks or it's slightly high for (SS) framework. Concerning $IM_2$, the (SS) framework is consistently better than that (U) framework. The improvement in $IM_1$ and $IM_2$ compared to baseline unsupervised frameworks is not drastically high, however, these results are still important because the marginal improvement in interpretability is happening simultaneously with consistent improvement in sparsity. Sparse counterfactual explanations always run the risk of not belonging to the data distribution of the target class.

From figure 2c it is evident that a sparse perturbation to the feature space can alter the class outputs either way Hence, a sparse counterfactual should have reduced interpretability, but on the contrary, our interpretability results have improved, although marginally. Now we present some individual counterfactual explanations obtained during our experimentation. The feature changes in the counterfactual explanation generated using (SS) framework generally sparser compared to the feature

|   |    | Month | Credit Amount | Installment % | Purpose | Decision |
|---|----|-------|---------------|---------------|---------|----------|
| 1 | U  | $15 \to 20$ | $1778 \to 2438$ | $2 \to 2.2$ | $--$ | high→low |
|   | SS | $15 \to 20$ | $1778 \to 2465$ | $--$ | $--$ | |
| 2 | U  | $16 \to 21$ | $--$ | $3 \to 4$ | $--$ | high→low |
|   | SS | $--$ | $3050 \to 3758$ | $--$ | A44→A48 | |

Table 2: Counterfactual generated on German Credit dataset.

changes involved in the counterfactual explanations generated using the (U) framework. Few examples are showed in the table 2 and table 3. Consider the second instance of the counterfactual query from the German Credit data-set, (SS) framework suggests: only by changing the loan application **purpose** from "Domestic Appliances" (A44) to "Retraining" (A48) the credit risk can go down significantly. This is associated with a corresponding change in the **credit amount**. The results obtained from the counterfactual of COMPAS data-set show evidence that (SS) framework is capturing the implicit correlation between various features. As an example in the second counterfactual instance (table 3) two features, age, and age-category are simultaneously changed in the counterfactual explanation produced using semi-supervised embedding, while instances from unsupervised embedding failed to capture such relation. Contrary to the opinion that sparser counterfactual explanations may neglect to change other correlated variables, (SS) framework makes sure it captures them, thus remaining within target class distribution. The sparser counterfactual explanations generation also indicates that the model can mine out fewer features that maximally influence the classifier decision.

|   |    | Age | Age category | Prior | Charge | Recidivism |
|---|----|-----|--------------|-------|--------|------------|
| 1 | U  | $--$ | $25 - 45 \to \geq 45$ | $2 \to 5$ | $--$ | $No \to Yes$ |
|   | SS | $--$ | $--$ | $2 \to 5$ | $--$ | |
| 2 | U  | $24 \to 48$ | $--$ | $0 \to 7$ | DL revoked $\to$ Robbery | $Yes \to No$ |
|   | SS | $24 \to 38$ | $\leq 25 \to 25 - 45$ | $0 \to 12$ | $--$ | |

Table 3: Counterfactual instances for COMPAS data-set.

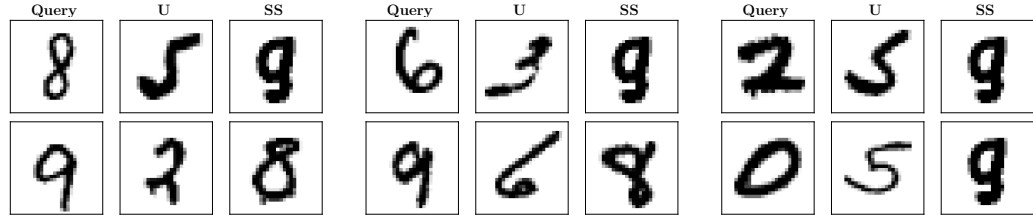

Figure 3: Comparing decoded images of `proto` embeddings generated through unsupervised and semi-supervised frameworks.

Further, we have defined an odd-even digit classification task on the MNIST dataset. In this case, embeddings obtained in a semi-supervised fashion show clear class separation as in figure 2d. We show the decoded images of `proto` embeddings generated through (U) and (SS) frameworks in figure 3. It can be observed that the decoded `proto` values for the (SS) framework are robust, because for all the three even queries in 3, the digit 9 happens to be the decoded `proto`. The primary reason being: convergence of prototype guided search to the class center in the embedding space. The well-formed digits are positioned around the class centers, while, the visually ambiguous digits are at the class boundaries. Thus the resulting `proto` is much stable in the (SS) framework. Embedding generated using an unsupervised framework does not show any clear class separation, hence the resulting prototypes are often ill-formed digits, and produces varied decoded `proto` images, depending on the start query.

## 6 CONCLUSION AND FUTURE WORK

We have empirically demonstrated semi-supervised embedding produces sparse counterfactual explanations. Sparse counterfactual explanations run the risk of not belonging to the data distribution of the target class. However, semi-supervised embedding ensures that that guided prototype lies within the target cluster with certain "robustness". For future work, we will explore the potential of a more rich data representation using semi-supervised Variation Auto Encoders (VAE) and imposing causality and feasibility constraints to derive a more faithful data embeddings.

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
