# OpenReview forum: "Semi-supervised counterfactual explanations"
_ICLR.cc/2021/Conference — Reject_

### Official Review · AnonReviewer1 · 2020-10-25
**This paper proposes to obtain counterfactual explanations. Over previous work, the goal is to improve the quality of counterfactuals by ensuring they are close to the distribution of the target subclass. In order to ensure the counterfactuals are from the target class manifold, an auto-encoder scheme is proposed. The auto-encoder is trained by encouraging its latent space to be discriminative of the target label.**

**Rating:** 4
**Confidence:** 5

**Review:**

My main concern for this paper is around several factors that I will list from the most important to minor.

i) It is unclear to me when and how generator training should be biased according to the class label, which the authors do by encouraging the latent space to be predictive of the class/target label. I have a few questions regarding this:
 a) Why is it insufficient to train a conditional version of a VAE to create separation in the latent space that guarantees the counterfactual is appropriately separated and lies on the manifold of the target class.

b) This specifically concerns the authors' claim that in the case of post-hoc training, input-output pairs of the model can be used to train the semi-supervised auto-encoder. Now this question is fundamental to me in terms of the goals of the counterfactual explanation. Is the goal to provide explanations in ways that allow us to identify issues with the classifier? Or is the goal of the counterfactual explanation to provide recourse to the end user. In the first case, I would rather see counterfactual explanations that are not in any way or form biased by the generative model, and hoping there is a perfect generative model that captures the entire distribution of the samples this model is supposed to generalize on. Now generating counterfactuals with such a perfect generator might actually give some hope of identifying issues worth debugging in a classifier like a bias toward specific classes etc. If the goal of the counterfactual explanation is to provide recourse, then simply hoping its a likely prototype from the target class is most definitely insufficient. Even if we assumed that it is in-fact sufficient, the post-hoc training on input-output pairs of the classifier will in fact bias the generator heavily and really does not reveal much about the classifier at all. The authors need to strongly consider and clarify what the goal is of their counterfactual explanation, what kinds of biases different methods induce in their explanation, why a conditional generator is not sufficient and potential harms of a jointly trained model.

ii) This brings me to my second question around empirical evaluation.
a) Include a conditional baseline
b) Compare to all the methods that train generator separately, jointly, and semi-supervised model.
c) what happens if the support of the training data for the generative model is mismatched from the data used for training.
d) Should counterfactual explanations actually be designed to provide fidelity to the classifier behavior or just be purely interpretable but not of great utility in actually improving the classifier. If the goal is to not improve the classifier but to audit it, I strongly feel simply providing qualitative summaries of sparsity and measures of interpretability in terms of target class probability are insufficient. They are also insufficient to provide recourse.

All of these factors significantly diminish this contribution. I recommend the authors give critical thought to the goals of counterfactual explanations and the effects of joint vs disjoint training in addition to merely semi-supervised and unsupervised training of the generative model, here an autoencoder.

---

### Official Review · AnonReviewer3 · 2020-10-27
**Semi-supervised counterfactual explanations**

**Rating:** 4
**Confidence:** 5

**Review:**

Given a classifier, this paper proposes to extend the use of auto-encoders to generate interpretable counterfactual explanations by building a latent space that is aware of the class tag information through joint training of the auto-encoder and the classifier. A counterfactual explanation is defined as more interpretable if it lies within or close to the data distribution of the training data of the black box classifier.
The experimental part of the paper tests the proposal by using different datasets and some metrics that are adapted to their definition. Experiments are well designed and well performed.

Strong points:
+ Generating interpretable counterfactual explanations is an open problem that is not only interesting from a theoretical point of view, but also in practical terms.
+ The paper describes in a fair way alternative methods and correctly identifies their strong and weak points.
+ The use of a semi-supervised autoencoder seems to be a good ingredient to get explanations that are close to the data distribution of the classifier.

Weak points:
- The proposed method (implementing a semi-supervised autoencoder) is simple from a technical point of view and does not represent any challenge.
- The paper proposes the following definition: "A counterfactual explanation is more interpretable if it lies within or close to the data distribution of the training data of the black box classifier." This definition of "interpretability" is too simple and it requires some discussion to be accepted.  This definition is not based on causal reasoning and because of this, we cannot assume that this kind of explanation is actionable or even feasible. Moreover, this definition is in contradiction with a causal view (a view that is fairly explained in the paper): counterfactuals are realizations from a probability distribution that is different from that of the classifier.  Lying within or close to the data distribution of the training data of the black box classifier is a property that needs justification.

This last point is not saying that to lie within or close to the data distribution of the training data is an undesired property for counterfactual explanations, but I miss a sounder motivation. I would suggest some improvements in this line:
+ I do not think that the concept of "interpretability" can be defined independently of a causal model of the world that limits what is feasible and what is not. For this reason, in order to check if the proposed methods represent a clear step towards "interpretability, a comparison with causal-based counterfactuals is necessary.
+ Testing counterfactuals is not possible because of the impossibility of ground truth data, but alternative approaches exist, mainly based on some kind of human evaluation. I am missing some experiments in this line.

Minor points:
- To have a more complete overview of the topic I would recommend to include some papers related to the concept of "algorithmic recourse", which identify some limitation of the counterfactual explanation approach.

---

### Official Review · AnonReviewer2 · 2020-10-28
**Simple method to address challenges with counterfactual explanations but there are questions surrounding evaluation metrics**

**Rating:** 6
**Confidence:** 3

**Review:**

Summary:
This paper continues an emerging line of research to find interpretable (post-hoc) counterfactual explanations of classifier predictions. While prior work has made advances in ensuring that resulting counterfactuals lie in the same data distribution as the original dataset by using auto-encoders, this paper provides a semi-supervised approach in an attempt to also ensure that they lie in the data distribution corresponding to the counterfactual label. The authors show the benefits of their approach on six datasets:  German Credit data, Adult Census data, MNIST, COMPAS, PIMA, and Breast Cancer Wisconsin data.

Pros:
- The paper builds on an interesting and emerging line of research and addresses an important conceptual problem faced by prior work.
- The authors present comprehensive experiments to show that their method leads to improved results on multiple datasets using multiple metrics, and accompany it with interesting qualitative analysis that further supports the quantitative results. The simplicity of the method would allow practitioners to generalize it to various other datasets.
- Except for a few issues, the paper is well written and easy to follow

Cons:
- The results are not uniformly in favor of the proposed method when compared to the unsupervised approach applied by prior work, and to the authors’ credit, they acknowledge that in the text.
- I would like to see more detail in discussion surrounding qualitative results, particularly on MNIST. Furthermore, in Tables 2 and 3, the authors share limited examples, it would be great to see more qualitative analysis in the rebuttal. I also didn’t quite understand what the authors mean by if the digit 9 happens to be the decoded proto for all three even queries presented in Figure 3, that the proto values for the framework are “robust”. Robust in what sense and to what exactly?
- The proximity metrics aren't very clear and there appears to be some discrepancy. The definitions of continuous and categorical proximity in this paper are different than the ones presented in Mothilal et al. (2020). If we go by their definitions (as the authors claim they do --- Section 5.3, first two lines), one would arrive at the definitions in this paper if we assumed that there is only one categorical variable and one continuous variable, but that is not the case in the datasets used in this paper.

Additional Questions:
- Please clarify this from Section 5.4 where you say, “(SS) framework fares better even for categorical proximity and continuous proximity metrics across all datasets except COMPAS dataset, where continuous proximity of (SS) framework is higher.” Based on my interpretation of Table 1, it appears that the performance on COMPAS dataset follows the same trend as on German Credit or Adult Income.

References:
The paper addresses an important question but at the same time misses on related work done in non-ML fields including Philosophy and Psychology. I recommend that the authors consider discussing some of the works on counterfactual selection from those fields in their related work section. Here are two for a start:
- Fazelpour, S. Norms in Counterfactual Selection. Philosophy and Phenomenological Research.
- Byrne, R. M. (2016). Counterfactual thought. Annual review of psychology, 67, 135-157.

---

### Official Review · AnonReviewer4 · 2020-10-28

**Rating:** 5
**Confidence:** 4

**Review:**

This paper presents a new approach for generating counterfactual explanations. Specifically, the presented method optimizing for a counterfactual explanation using a weighted loss function of L_pred, L_sparsity, L_recon, and L_proto, and differs from previous works in the manner in which the latter two losses are computed. In more detail, whereas prior work computes L_recon and L_proto using the reconstruction and latent space distance of *unsupervised* models, the presented method computes these losses using a semi-supervised setup whereby a model is jointly trained to minimize reconstruction and class-conditional loss. The authors conduct experiments on a number of real- and mixed-valued datasets, which is welcome in a field where broad experimentation is historically lacking.

Motivation:
Furthermore, I believe the work would benefit from a stronger motivation section and justification for the approach. Specifically, why would one not achieve the desired result -- i.e., of ensuring that the CFE is of the target class -- by simply increasing the weight of L_pred. It is peculiar for this not to happen, as counterfactual explanation must be, by definition, from another (desirable) class, and so it seems natural to expect this.

References:
The authors have missed a number of key citations. Specifically, such terms as "Actions", "Actionability", "Feasibility", and "Interpretability" are under-defined, perhaps the authors can refer to a recent survey paper [1] to clarify this terminology. Moreover, in the introduction, the authors suggest that counterfactual explanations are being used for practical applications, and/or for bias reduction, both without citations. The related work section also lacks such relevant citations as [2-5]. Sec 4 describes the "semi-supervised auto-encoder" without citing relevant work (as a non-expert in DL architectures, I may be wrong here) [6,7]. Finally, arguments such as  "interpretability will improve the trust among data subjects" could be made more concrete if one cited works as [8-11].

Other comments and nits;
- [sec 1; par 1] "provide necessary actions to receive a more favorable decision" is a common misconception in the CFE literature; this was shown not to be the case in [5,12]. perhaps replace with "provides meaningful explanations as to what features would receive favorable treatment" (not actions).
- [sec 1; par 2] it is unclear how "data distr. of the model" & "data distr. of individuals" are different?
- [sec 2; par 3] "input-output pair dat of the black-box" is unclear where the inputs are sampled from?
- [sec 3; par 2] the statement "they may not be interpretable" is unclear, because (as alluded above) the term interpretable is undefined and one would have to read Van Looveren to understand the meaning here.
- [sec 3; par 3] x_0 undefined.
- [sec 3; par 3] L_proto should be L_proto^\mathcal{X} as above.
- [sec 3; par 4] are c, gamma, and theta tuned for each "data set" or "data instance"?
- [sec 3; par 4] "Ltd" reference undefined.
- [sec 4.2; par 2] "clearly the separation is due" is quite strong of a claim; perhaps the training procedure for one model was different? Do different random initializations result in similar trends? Although this observation does not relate to the primary purpose of the paper (which is to then use such distributions in another optimization problem for CFE generation), it may be better to soften such strong claims.
- [sec 5; par 1] "reverse lookup" undefined
- [sec 5.2; par 1] what criteria were used to select the weights in the training loss for the semi-supervised setting? And how does training affect the resulting CFE?
- [sec 5.3; par 2] does the cat-proximity loss need "1 - "? Are we not aiming to minimize this distance as in cont-proximity? Perhaps I am mistaken.

Refs:
- [1] Karimi et al., https://arxiv.org/pdf/2010.04050.pdf
- [2] Guidotti et al. https://arxiv.org/pdf/1805.10820.pdf
- [3] Sharma et al., https://arxiv.org/pdf/1905.07857.pdf
- [4] Laugel et al., https://arxiv.org/pdf/1712.08443.pdf
- [5] Karimi et al., https://arxiv.org/pdf/2002.06278.pdf
- [6] Oza and Patel, https://arxiv.org/pdf/1904.01198.pdf
- [7] Rudy and Taylor, https://arxiv.org/pdf/1412.7009.pdf
- [8] Fjeld et al., https://papers.ssrn.com/sol3/papers.cfm?abstract_id=3518482
- [9] Lipton, https://arxiv.org/abs/1606.03490
- [10] Doshi-Velez and Kim, https://arxiv.org/abs/1702.08608
- [11] Doshi-Velez et al., https://arxiv.org/pdf/1711.01134.pdf
- [12] Karimi et al., https://arxiv.org/pdf/2006.06831.pdf

---

### Decision · Program_Chairs · 2021-01-07
**Final Decision**

**Decision:**

Reject

**Comment:**

There is a general consensus on the fact that the paper is not yet ready for publication. I encourage the authors to carefully address the detailed concerns raised by the reviewers, which include  among other: i) the incompleteness of the literature overview, which should include the references provided by the reviewers, ii) poor (or bias towards the proposed approach) experimental evaluation, and iii)  a vague treatment of key terms in the interpretability literature  like feasibility (e.g., to make sure that the counterfactual lie in high data density regions).